# Natural boundaries for scattering amplitudes

**Sebastian Mizera**[⋆]

Institute for Advanced Study, Einstein Drive, Princeton, NJ 08540, USA

⋆ smizera@ias.edu

## Abstract

Singularities, such as poles and branch points, play a crucial role in investigating the analytic properties of scattering amplitudes that inform new computational techniques. In this note, we point out that scattering amplitudes can also have another class of singularities called *natural boundaries* of analyticity. They create a barrier beyond which analytic continuation cannot be performed. More concretely, we use unitarity to show that $2 \to 2$ scattering amplitudes in theories with a mass gap can have a natural boundary on the second sheet of the lightest threshold cut. There, an infinite number of ladder-type Landau singularities densely accumulates on the real axis in the center-of-mass energy plane. We argue that natural boundaries are generic features of higher-multiplicity scattering amplitudes in gapped theories.

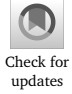

## 1  Introduction

Physical behavior of scattering amplitudes is encoded in their analytic properties. One does not need to search hard to find numerous examples where understanding of the analytic structure lead to significant improvements in computational techniques [1–6]. There is therefore great interest in further exploration of the analytic properties of scattering amplitudes, especially those general enough to be applicable to realistic processes in the Standard Model and beyond.

What kind of singularities are known to appear in scattering amplitudes? At tree-level in perturbation theory, we only encounter poles of the type $(s - m^2)^{-1}$, together with their multi-particle generalizations. At low loop level, direct computations give square-root and logarithmic branch points of the kind $f^{a/2} \log^b f$ for some integers $a, b$ and a function $f$ of the kinematics [7–11]. It is also known that scattering amplitudes can have branching of arbitrary order, for instance, $s^{\alpha(t)}$ in the Regge limit (where $s$ is large and $\alpha(t)$ is the Regge trajectory) [12, 13] or $s^\epsilon$ in dimensional regularization around $4 - 2\epsilon$ dimensions [14]. Theories of quantum gravity are expected to produce essential singularities of the form $e^{-s}$ in the high-energy limit $s \to \infty$ at fixed angle [15, 16]. Resummation of diagrams can lead to non-holonomic singularities of the form $(1 - f^{a/2} \log^b f)^{-1}$ [17–19]. It is also known that scattering amplitudes can have distributional support in special limits, such as deep inelastic scattering in QCD [20], or accumulation curves of singularities in unphysical kinematics [21]. One may ask if the list of possible singularities has been exhausted.

Mathematically, analytic functions can certainly have another feature called a *natural boundary* of analyticity, formed by a dense set of singularities. As such, they provide barriers to analytic continuation. Natural boundaries are no strangers to physicists; for example, elliptic functions such as the Dedekind eta function $\eta(\tau)$ have a singularity at every rational $\tau$ and hence a natural boundary on the real $\tau$-axis, see, e.g., [22]. In Sec. 2.1 we review the basic mechanism behind such accumulations of singularities. A natural question arises, whether scattering amplitudes can have natural boundaries and where to find them.[1]

To answer this question, we turn to the simplest non-analytic feature of the $2 \to 2$ S-matrix: the branch cut for the lightest-threshold exchange, starting at $s = 4m^2$, where $s$ is the center-of-mass energy squared. We take the mass $m$ to be strictly positive. In four space-time dimensions, the threshold is square-root branched, meaning that it divides the $s$-plane into two sheets glued by a branch cut, see Fig. 1. By convention, they are called the *first* (or *physical*) and *second* sheet respectively. We will denote the connected part of the S-matrix on the $i$-th sheet $T_i(s, z)$, where $z = \cos\theta$ is the cosine of the scattering angle and is held fixed (the analytic structure of the S-matrix at fixed momentum transfer $t$ is less understood, see App. A). It is on the second sheet that we will encounter a natural boundary, see Fig. 1 (right).

The physical S-matrix can be recovered by approaching the branch cut from the upper half-plane on the first sheet or lower half-plane on the second. We would like to emphasize that the second sheet is arguably the one carrying *more* physically-interesting analytic features.

---

[1]In order to avoid circular arguments, we assume that the spectrum of particles in the theory itself is not dense and it does not have accumulation points. In most of this work we will consider a theory with a single scalar of mass $m > 0$. Note that scattering amplitudes in superstring theory have a natural boundary in the (non-holomorphic) coupling constant (see, e.g., [23]), but it is not a kinematic singularity.

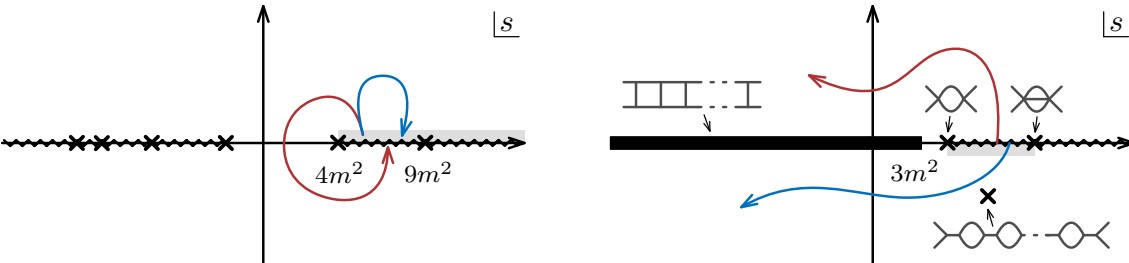

Figure 1: Two sheets of the lightest threshold cut at $s > 4m^2$, in the $s$-plane at fixed $z$, on which $T_1(s,z)$ and $T_2(s,z)$ are defined respectively. **Left:** First sheet has the $s$-channel normal-threshold cuts extending to the right, while the $t,u$-cuts run to the left. The physical region $s > 4m^2$ is approached from the upper half-plane (orange). Two paths of analytic continuation (red and blue) go through the elastic region $(4m^2, 9m^2)$ and end up on the second sheet. **Right:** Second sheet has a natural boundary (thick black line) extending throughout $s \leqslant 3m^2$ and arising from a dense accumulation of ladder-type Landau singularities. Additional singularities, such as bound-state poles in the lower half-plane might exist, but we illustrate them only schematically. The physical region (gray) is approached from the lower half-plane, from which two paths of analytic continuation (red and blue) emerge. The S-matrix cannot be analytically continued past the natural boundary.

For instance, it contains the resonance poles of unstable particles, resulting in the Breit–Wigner peaks observable in particle colliders. For this reason, we believe that properties of $T_2(s,z)$ deserve to be understood at a deeper level. For example, one of the questions we raise is whether the existence of the natural boundary can result in quantifiable effects on the physical region. Another motivation is to learn to what extent the properties of $T_2(s,z)$ can be used as an input in the non-perturbative S-matrix bootstrap [24].

A connection between the two sheets is provided by unitarity, which embodies the physical principle of probability conservation, or—more specifically—its analytic continuation [25–35]. To be concrete, in four dimensions the $T_i$'s are related by

$$T_1(s,z) - T_2(s,z) = \frac{\sqrt{4m^2 - s}}{\sqrt{s}} \int\limits_{P>0} \frac{dz_1\, dz_2}{\sqrt{P(z; z_1, z_2)}}\, T_1(s, z_1)\, T_2(s, z_2), \tag{1}$$

where $P = 1 - z^2 - z_1^2 - z_2^2 + 2z z_1 z_2$. The left-hand side is the analytic continuation of the imaginary part of the amplitude and the right-hand side corresponds to the continuation of unitarity cuts. As such, (1) provides a consistency condition for $T_2$ if $T_1$, or at least some of its properties, are known. One can write a formal solution for $T_2$ in terms of holomorphic unitarity cuts [36]. We review these aspects in Sec. 2.2-2.3.

Using unitarity, we will show that $T_2(s,z)$ has to have an infinite number of singularities at the non-perturbative level, corresponding to intermediate states going on-shell and being interpretable as classical scattering trajectories in complexified space-time. In perturbation theory, these are known as *anomalous thresholds* or *Landau singularities* [37–39].

To demonstrate the existence of a natural boundary, we consider specific singularities of the ladder-type, see Fig. 2. They correspond to scattering of two particles with $n$ mediation points, at which $k_i$ particles are being exchanged. We show that such a singularity develops whenever the scattering angle reaches the value $\theta = \theta^*_{k_1, k_2, \ldots, k_n}$, which in the simplest case reads

$$\theta^*_{k_1, k_2, \ldots, k_n} = \sum_{i=1}^{n} \theta^t_{k_i}(s), \qquad \text{with} \qquad \theta^t_{k_i}(s) = \arccos\left(1 + \frac{2k_i^2 m^2}{s - 4m^2}\right). \tag{2}$$

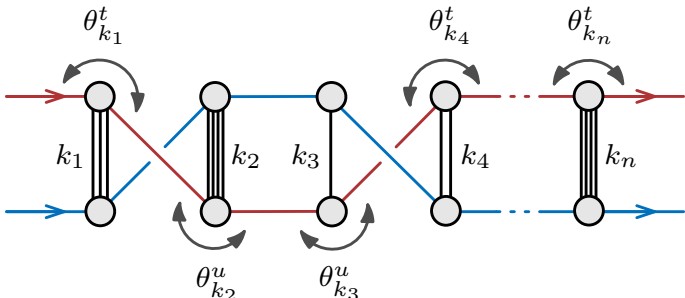

Figure 2: The class of ladder-type Landau singularities building up the natural boundary. Every edge is an on-shell state in the theory. The resulting singularity contributes at the specific angle given by (2), and more generally by (39), with $\theta^u_{k_i} = \theta^t_{k_i} + \pi$. Each of the $n$ rungs exchanges $k_i \geqslant 1$ particles.

Each $\theta^t_{k_i}(s)$ is the specific scattering angle across the interaction point with $k_i$ states. In order to find the singular angle $\theta$, one simply adds them up modulo $2\pi$. A more general case is derived in Sec. 3.2. Note that positions of these singularities in the Mandelstam invariants $(s, t)$ would be extremely difficult to describe: it is working with the scattering angles that allows us to write such a compact expression for any $n$. In contrast with the perturbation-theory Landau analysis, which only gives a necessary condition for singularities, unitarity is a much stronger constraint which also guarantees sufficiency, up to possible cancellations.

Natural boundaries arise from a dense accumulation of such singularities as $n \to \infty$, even if we consider a small subset with all $k_i$'s equal to $k$, i.e. equal-rung ladders. In Sec. 3.3, we analyze positions of these singularities in the $s$-plane for fixed-$\theta$ and fixed-$t$ scattering. As a matter of fact, they decompose into sectors depending on how many "windings" it takes to construct the whole diagram by adding individual angles according to (2), which could be anywhere between 0 and $n-1$. Each of them contributes a single singularity, such as a branch point, in the $s$-plane. The simplest case is that of the fixed physical (real) angle $\theta$, for which a natural boundary is located on the half-line $s \leqslant (4 - k^2)m^2$, as illustrated in Fig. 1 (right) for $k = 1$.

The natural boundary should be thought of as an effect of resumming over ladder-type diagrams, and in that sense, it would not be visible at any finite order in perturbation theory. It is a feature similar to the 1PI resummation of propagators, which moves positions of one-particle poles to the complex $s$-plane, even though no individual Feynman diagram has such a pole.

We also consider two situations under which we see that singularities densely accumulate on the half-line $s \leqslant (4 - k^2)m^2$, but are otherwise more spread-out in the complex plane, illustrated later in Fig. 3.3. This happens either when the scattering angle $\theta$ is complex or if we work at fixed-$t$ away from forward limits. In Sec. 4 we give evidence that natural boundaries will be generically present in higher-multiplicity scattering amplitudes in gapped quantum field theories.

**Outline.** This work is structured as follows. In Sec. 2 we review the necessary background material on lacunary functions and analytic continuation of elastic unitarity. In Sec. 3 we analyze singularities on the second sheet of the lightest-threshold branch cut and explain how a natural boundary is formed. We close in Sec. 4 with a discussion of the results and open problems. In App. A and B we review the results on upper half-plane analyticity at fixed momentum transfer and scattering angle.

**Note added.** During the write-up stage of this work, we found Refs. [40–43] which previously discussed a possibility of natural boundaries on unphysical sheets.

## 2 Background

In this section, we review some background material on lacunary functions and elastic unitarity.

### 2.1 Lacunary functions

In this section, we explain how to see natural boundaries forming on one of the classic examples. Consider the function $f(z)$ with the following Taylor expansion around the origin:

$$f(z) = z + z^2 + z^4 + z^8 + z^{16} + \ldots = \sum_{k=0}^{\infty} z^{2^k}. \tag{3}$$

This series converges for $|z| < 1$ and hence $f(z)$ is analytic in the unit disk. Since $f(1) = 1 + 1 + 1 + \ldots$, the function is singular at $z = 1$ and has a unit radius of convergence. Let us investigate other places on the circle $|z| = 1$ where singularities can arise. Using the above definition, $f(z)$ satisfies the relations

$$f(z) = z + f(z^2) \tag{4}$$
$$= z + z^2 + f(z^4) \tag{5}$$
$$= z + z^2 + z^4 + f(z^8) \tag{6}$$
$$= \ldots. \tag{7}$$

The first equality evaluated at $z = -1$ gives $f(-1) = -1 + f(1)$, so we find that $f(-1)$ is singular. By similar logic, the second equality evaluated at $z = \pm i$ implies that also $f(\pm i) = \pm i - 1 + f(1)$ diverges. Repeating this exercise, one finds that every root of unity of the form $z^{2^k} = 1$ is in fact singular for every $k = 0, 1, 2, \ldots$. These singularities therefore densely cover the unit circle (meaning that every point on $|z| = 1$ is either singular or arbitrarily-close to a singularity), thus creating a natural boundary of analyticity. This boundary prevents analytic continuation of $f(z)$ beyond the unit disk.

The presence of the natural boundary is closely tied to the large gaps (lacunae) in the Taylor series (3), i.e. the absence of the terms between every $z^{2^k}$ and $z^{2^{k+1}}$. One can ask how to diagnose if a natural boundary exists. Let us start with a Taylor series

$$g(z) = \sum_{k=0}^{\infty} a_k z^{\lambda_k}, \tag{8}$$

with a unit radius of convergence. Here all $a_k$ are non-zero and the question is how much do gaps in the increasing sequence $(\lambda_0, \lambda_1, \lambda_2, \ldots)$ have to grow. Hadamard's gap theorem states that if $\lambda_k$ grows as $\delta^k$ when $k \to \infty$ for any $\delta > 1$, then $g(z)$ has a natural boundary. Such a function is called *lacunary*. Above, we have seen an example with $\lambda_k = 2^k$. This theorem does not give a necessary condition and can be improved in various ways; for example, Fabry's gap theorem shows it is enough that $\lambda_k/k \to \infty$ as $k$ becomes large for $g(z)$ to develop a natural boundary. For more details on lacunary functions, we refer to [44] and references therein.

### 2.2 Analytic continuation of elastic unitarity

In this section we review the derivation of the elastic unitarity equation. For simplicity, we treat a scalar theory with a single particle of mass $m$. In this case, the connected part of a $2 \to 2$ amplitude $T(p_1, p_2; p_3, p_4) = T(s, z)$ can be expressed in terms of the center of mass energy squared, $s = (p_1 + p_2)^2$, and the cosine of the scattering angle $\theta$,

$$z = \cos\theta = 1 + \frac{2t}{s - 4m^2}, \tag{9}$$

with the remaining Mandelstam invariants given by $t = (p_2 - p_3)^2$ and $u = (p_1 - p_3)^2 = 4m^2 - s - t$.

At fixed physical $z \in (-1, 1)$, the amplitude has the $s$-channel branch cuts when $s > (km)^2$, as well as $t$- and $u$-channel branch cuts when $s < m^2(4 - \frac{2k^2}{1 \mp z})$ (translating to $t, u > k^2 m^2$) on the physical sheet for $k = 2, 3, \ldots$. The Euclidean region is the interval $s \in (4m^2 \frac{z-1}{z+1}, 4m^2)$ between them, where the amplitude is analytic and real, aside from one-particle poles whose positions are determined by plugging in $k = 1$ above. The half-line $s > 4m^2$ with $z \in (-1, 1)$ is called the $s$-channel physical region because it corresponds to the kinematics with real scattering angles and energies. It is known that $T(s, z)$ is analytic in the complex $s$-plane minus the branch cuts, at least to all orders in perturbation theory, see App. B.

Elastic unitarity relates the imaginary part of $T(s, z)$ to its unitarity cuts in the region $s \in (4m^2, 9m^2)$, where only two intermediate particles can be exchanged in the $s$-channel. From now on, we will call it the *elastic region*. What is important to us is that we can express the imaginary part as a difference between $T(s, z)$ on the first and second sheet of the $s > 4m^2$ branch cut. To make this precise, let us rename $T(s, z)$ evaluated on the first sheet as $T_1(s, z)$. In particular, the scattering amplitude in the physical region is given by the boundary value

$$T_1(s, z) = \lim_{\varepsilon \to 0^+} T(s + i\varepsilon, z). \tag{10}$$

Similarly, the amplitude on the second sheet $T_2(s, z)$ is defined by looping around the $s = 4m^2$ branch point once. We will see below that in even space-time dimensions, there are only two sheets of the lightest-threshold branch cut, which means we can analytically continue either in a clockwise or anticlockwise direction and land on the same sheet. When evaluated in the elastic region, we have

$$T_2(s, z) = \lim_{\varepsilon \to 0^+} T(s - i\varepsilon, z) = \lim_{\varepsilon \to 0^+} \overline{T(s + i\varepsilon, z)}. \tag{11}$$

The first equality simply means we can approach the second sheet by going underneath the $s > 4m^2$ branch cut, as long as $s < 9m^2$. The second equality is implied by the Schwarz reflection principle in the presence of the Euclidean region. We thus have

$$\operatorname{Im} T(s, z) = \frac{1}{2i}\left(T_1(s, z) - T_2(s, z)\right) \tag{12}$$

in the elastic region. It is a non-trivial result connecting unitarity and analyticity, because it is not always guaranteed that the imaginary part equals the discontinuity [36, Sec. 4.4]. The discussion is unchanged if we worked at fixed momentum transfer $t \in (-4m^2, 0)$ instead of the fixed angle $z$.

Recall that unitarity is the operator statement $SS^\dagger = \mathbb{1}$ on the full S-matrix. After expanding $S = \mathbb{1} + iT$ and acting with 2-particle in and out states, we find the standard result

$$\frac{1}{2i}\left(T_1(s, z) - T_2(s, z)\right) = 4 \int d^D\ell \, \delta^+[\ell^2 - m^2] \, \delta^+[(p_1 + p_2 - \ell)^2 - m^2] \tag{13}$$

$$\times T_1(p_1, p_2; \ell, p_1 + p_2 - \ell) \, T_2(p_1 + p_2 - \ell, \ell; p_3, p_4)$$

in the elastic region. While the correct normalization of the right-hand side is fixed by unitarity, it is immaterial to our discussion and hence we set it to 4 for later convenience. The loop integrand depends on $T_{1/2}$ expressed in terms of the Lorentz vectors of the external and internal states. It also involves the delta functions $\delta^+(q^2 - m^2) = \theta(q^0)\delta(q^2 - m^2)$ putting the two intermediate particles on-shell with positive energy in the mostly-minus signature. The unitarity equation (13) is illustrated diagrammatically in Fig. 3.

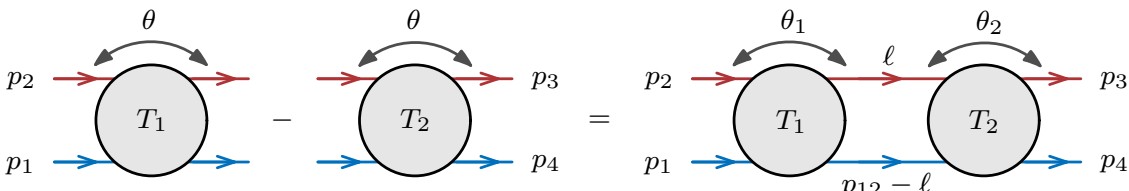

Figure 3: Diagrammatic summary of the analytically-continued unitarity equation (1), and more generally (22), providing a relation between $T_1$ and $T_2$.

The goal is to express the right-hand side in terms of $T_1(s,z_1)$ and $T_2(s,z_2)$ convolved with some integral over the cosines of angles

$$z_1 = \cos\theta_1 = 1 + \frac{2(p_2-\ell)^2}{s-4m^2}, \qquad z_2 = \cos\theta_2 = 1 + \frac{2(\ell-p_3)^2}{s-4m^2}. \tag{14}$$

The total energy $s$ is shared between all the amplitudes appearing above, as is clear from Fig. 3. We follow the derivation outlined in [45, Sec. 3.3.2].

The first step is to split the loop momentum $\ell = \ell_\parallel + \ell_\perp$ into the 3-dimensional component $\ell_\parallel$ in the directions generated by the external momenta $p_1, p_2, p_3$ (recall that $p_4$ is not independent because of the momentum conservation) and the $(D-3)$-dimensional orthogonal complement $\ell_\perp$. The signature of the two components is $(+--)$ and $(--\ldots)$ respectively. We take $D > 3$. The point is that the loop momentum only enters through the Lorentz-invariant combinations $\ell \cdot p_i = \ell_\parallel \cdot p_i$ and $\ell^2 = \ell_\parallel^2 + \ell_\perp^2$, so we can trade

$$\int d^3\ell_\parallel = \frac{1}{\sqrt{\det \mathcal{G}_{p_1 p_2 p_3}}} \int d(\ell \cdot p_1)\,d(\ell \cdot p_2)\,d(\ell \cdot p_3), \tag{15}$$

where the Gram determinant of the external momenta in the square root evaluates to

$$\det \mathcal{G}_{p_1 p_2 p_3} = \det \begin{bmatrix} m^2 & \frac{s-2m^2}{2} & \frac{s+(s-4m^2)z}{4} \\ \frac{s-2m^2}{2} & m^2 & \frac{s-(s-4m^2)z}{4} \\ \frac{s+(s-4m^2)z}{4} & \frac{s-(s-4m^2)z}{4} & m^2 \end{bmatrix} = \tfrac{1}{16}s(s-4m^2)^2(1-z^2). \tag{16}$$

The dependence on $\ell_\perp$ almost entirely drops out, except for the modulus $\ell_\perp^2 < 0$. We can therefore go to the radial coordinates and integrate out the angular components to give

$$\int d^{D-3}\ell_\perp = \frac{\pi^{(D-3)/2}}{\Gamma(\frac{D-3}{2})} \int_{\ell_\perp^2 < 0} d(-\ell_\perp^2)(-\ell_\perp^2)^{\frac{D-5}{2}}. \tag{17}$$

Here, $-\ell_\perp^2$ can be expressed as

$$-\ell_\perp^2 = \ell_\parallel^2 - \ell^2 = -\frac{\det \mathcal{G}_{p_1 p_2 p_3 \ell}}{\det \mathcal{G}_{p_1 p_2 p_3}}. \tag{18}$$

Let us hold off evaluating it for a while. Note that $\ell_\perp^2 < 0$ is the only constraint on the integration contour.

At this stage, we are left with four integrations: over $\ell^2$ and three $\ell \cdot p_i$'s. We can localize two of them using the two delta functions putting two particles on the cut. Since we are already

imposing the $s$-channel kinematics, energies of the on-shell particles cannot be negative, i.e. $\delta^+$ can be simply replaced with the regular $\delta$ functions. On their support, we find

$$\ell^2 = m^2, \qquad \ell \cdot p_1 = \frac{s + z_1(s - 4m^2)}{4}, \tag{19}$$

$$\ell \cdot p_2 = \frac{s - z_1(s - 4m^2)}{4}, \qquad \ell \cdot p_3 = \frac{s - z_2(s - 4m^2)}{4}, \tag{20}$$

which gives

$$\det \mathcal{G}_{p_1 p_2 p_3 \ell} = -\frac{1}{64} s(s - 4m^2)^3 (1 - z^2 - z_1^2 - z_2^2 + 2 z z_1 z_2). \tag{21}$$

Localizing two of the variables and exchanging the other two for $(z_1, z_2)$ gives the final result:

$$T_1(s, z) - T_2(s, z) = c_D(z) \frac{(4m^2 - s)^{\frac{D-3}{2}}}{\sqrt{s}} \int\limits_{P > 0} \frac{\mathrm{d}z_1 \, \mathrm{d}z_2}{P(z; z_1, z_2)^{\frac{5-D}{2}}} T_1(s, z_1) T_2(s, z_2), \tag{22}$$

where

$$P(z; z_1, z_2) = 1 - z^2 - z_1^2 - z_2^2 + 2 z z_1 z_2, \tag{23}$$

and $c_D(z) = 2^{4-D} \pi^{\frac{D-3}{2}} (1 - z^2)^{\frac{4-D}{2}} / \Gamma(\frac{D-3}{2})$. In D = 4, we have $c_4(z) = 1$ and thus recover the advertised equation (1). The prefactor features normal-threshold singularity at $s = 4m^2$ and the pseudo-normal threshold (coinciding with the second-type singularity) at $s = 0$.

At this stage, we can analytically continue the expression (22) in $s$ or $z$. One point of view is to treat $T_1$ as an input and view (1) as an integral equation for $T_2$, subject to the constraints (10) and (11). The way we will make use of (1) is even simpler: given the known analytic properties of $T_1$, we will ask what singularities of $T_2$ are implied.

From (22) we can attempt to read off the nature of the branch point at $s = 4m^2$. The right-hand side has a square-root branch point in even space-time dimensions D and no branch point in odd D, if we were to assume that the integrand of (22) does not have additional zeros or singularities at the leading order around $s = 4m^2$ (recall that $m > 0$ and D > 3). For a $k$-particle cut with $k \geqslant 2$, a similar derivation gives the behavior $(k^2 m^2 - s)^{\frac{(k-1)D - k - 1}{2}}$ near the threshold $s = k^2 m^2$, see, e.g., [17, Sec. 2]. Since the right-hand side was computing the discontinuity of the original amplitude $T(s, z)$, it suggests the leading branching behavior is square-root $(k^2 m^2 - s)^{\frac{(k-1)D - k - 1}{2}}$ if $k$ and D are even and logarithmic $(k^2 m^2 - s)^{\frac{(k-1)D - k - 1}{2}} \log(k^2 m^2 - s)$ otherwise. This result matches the field-theory prediction based on Feynman diagrams [38, Sec. 4]. However, the assumption that the integrand is well-behaved around the threshold does not hold up to scrutiny in practical one-loop examples once spinning particles are included, leading to different threshold behavior, see, e.g., [45, Sec. 6.6]. Hence, we will not pursue estimating types of branch points further.

## 2.3 Holomorphic unitarity cuts

Starting from the unitarity equation (22), we can try to solve for $T_2$ entirely as a functional of $T_1$'s. Let us trivially rearrange this equation to the form

$$T_2(s, z) = T_1(s, z) - c_D(z) \frac{(4m^2 - s)^{\frac{D-3}{2}}}{\sqrt{s}} \int\limits_{P > 0} \frac{\mathrm{d}z_1 \, \mathrm{d}z_2}{P(z; z_1, z_2)^{\frac{5-D}{2}}} T_1(s, z_1) T_2(s, z_2). \tag{24}$$

We then repeatedly keep plugging the $T_2$ on the left-hand side into the right-hand side. For example, in the first step, we replace $T_2(s, z_2)$ on the right-hand side by plugging in the

expression (24), giving

$$T_2(s,z) = T_1(s,z) - c_D(z)\frac{(4m^2-s)^{\frac{D-3}{2}}}{\sqrt{s}} \int\limits_{P>0} \frac{dz_1\, dz_2}{P(z;z_1,z_2)^{\frac{5-D}{2}}} T_1(s,z_1)\, T_1(s,z_2) \tag{25}$$

$$+ c_D(z)\frac{(4m^2-s)^{D-3}}{s} \int\limits_{P>0} \frac{dz_1\, dz_2\, dz_3\, dz_4\, c_D(z_2)}{P(z;z_1,z_2)^{\frac{5-D}{2}} P(z_2;z_3,z_4)^{\frac{5-D}{2}}} T_1(s,z_1)\, T_1(s,z_3)\, T_2(s,z_4),$$

where the integration contour in the final line imposes that both $P$'s are positive. Notice that the dependence on $z_2$ enters only mildly through the $c_D(z_2)$ and the $P$'s and can in principle be integrated out, resulting in a new kernel (a Lauricella function or an elliptic function in D = 4) depending only on the physical variables $z$ and $z_1, z_3, z_4$. However, having a simpler integral expression will not be important in our applications and hence we leave $z_2$ unintegrated to make the formulae more compact.

At this stage, one can keep going and proceed with plugging in the expression (24) into itself over and over. This strategy results in the required formal expression for $T_2(s,z)$:

$$T_2(s,z) = T_1(s,z) + \sum_{c=1}^{\infty} (-1)^c \frac{(4m^2-s)^{\frac{(D-3)c}{2}}}{s^{\frac{c}{2}}} \tag{26}$$

$$\times \int\limits_{P>0} \frac{d^{2c}z \prod_{i=0}^{c-1} c_D(z_{2i})}{\prod_{i=0}^{c-1} P(z_{2i};z_{2i+1},z_{2i+2})^{\frac{5-D}{2}}} \prod_{i=0}^{c} T_1(s,z_{2i+1}).$$

For each term, we identify $z_0 = z$ and $z_{2c+1} = z_c$ and once again the integration contour imposes that all $P$'s appearing in the integrand are positive. All the $z_i$'s with even positive $i$ can be integrated out. The sum goes over $c$ cuts obtained by gluing together $c+1$ copies of $T_1$. We assume it converges. This way of joining the $T_1$ matrix elements is called a *holomorphic unitarity cut* [36]. It is closely related to Fredholm theory, which studies solutions of integral equations of the type (22), see, e.g., [46, Ch. 6].

## 2.4 Lehmann and Zimmermann ellipses

Let us mention one of the classic results in the S-matrix theory concerning the regions of analyticity of $T(s,z)$ and $\mathrm{Im}\, T(s,z)$ known as the *Lehmann ellipses* [47]. At any given $s$, they are the ellipses in the $z$-plane with foci at $z = \pm 1$ and the semi-major axes $e_1 > 1$ and $e_2 = 2e_1^2 - 1 > e_1$ respectively. Here, $e_i$ depend on the details of the theory and in our case are given by

$$e_1 = \sqrt{1 + 8m^2 \frac{|s| + \mathrm{Re}\, s}{|s-4m^2|^2}}, \qquad e_2 = 1 + 16m^2 \frac{|s| + \mathrm{Re}\, s}{|s-4m^2|^2}, \tag{27}$$

assuming Mandelstam analyticity (which postulates that $T$ is analytic on the physical sheet except for the branch cuts $s, t, u \geqslant 4m^2$ and possible bound-state poles for $0 \leqslant s, t, u < 4m^2$). The statement is that $T(s,z)$ and the analytic continuation of $\mathrm{Im}\, T(s,z)$ are analytic in the small and large ellipses respectively, except for possible poles. For example, the two-particle threshold in the $t$- and $u$-channels begin at $z = \pm\left(1 + \frac{8m^2}{s-4m^2}\right)$ for any $s$, which both lie on the boundary of the small ellipse. On the other hand, the large ellipse contains the above branch points, meaning that they cannot be singularities of the imaginary part, i.e. the right hand side of (22). For this singularity to cancel out on the left-hand side, it must be that $T_2(s,z)$ also has normal thresholds in the $t$- and $u$-channels. Alternatively, we could have concluded the same fact from the representation (26).

Zimmermann proved that analyticity of $\mathrm{Im}\,T(s,z)$ can be extended progressively to larger and larger ellipses with semi-major axes

$$e_n = \cosh\left(n\,\mathrm{Re}\,\mathrm{arccosh}\left(1+\frac{8m^2}{s-4m^2}\right)\right),\tag{28}$$

minus a number of normal- and anomalous-threshold branch cuts [28, Sec. 4.3]. Cases $n=1,2$ reduce to (27). In Sec. 3, we will reinterpret the corresponding branch points as creating a natural boundary for $T_2(s,z)$ as $n\to\infty$.

# 3 Natural boundaries on the second sheet

In this section we investigate the analytic properties of the amplitude on the second sheet of the lightest threshold cut guaranteed by unitarity and find a natural boundary. We specialize to $D=4$ for concreteness.

## 3.1 Unitarity and anomalous thresholds

We start by explaining how to use elastic unitarity equation (1) to determine positions of singularities in $z$ for a given $s$. The goal will not be to classify all of them, but instead focus only on the specific class of singularities relevant to our discussion.

The right-hand side of (1) features the integral

$$\int\limits_{P>0}\frac{\mathrm{d}z_1\,\mathrm{d}z_2}{\sqrt{P(z;z_1,z_2)}}\,T_1(s,z_1)\,T_2(s,z_2).\tag{29}$$

As we analytically continue this expression in $s$ and $z$, the contour of integration will keep getting deformed. To make it more concrete, let us fix $s$ and assume we already know that both $T_i(s,z_i)$ are singular at some specific values $z_i=z_i^*$, not necessarily corresponding to physical angles. The integral will then certainly become singular for $z=z^*$ satisfying

$$P(z^*;z_1^*,z_2^*)=0.\tag{30}$$

It is an example of an endpoint singularity, which cannot be avoided by deforming the integration contour. As mentioned before, the unitarity integral can have more complicated pinch and endpoint singularities too, in particular those independent of $z$, and we leave the full classification until future work. In the present case, (30) has the two solutions

$$z^* = z_1^* z_2^* \pm \sqrt{(1-z_1^{*2})(1-z_2^{*2})}.\tag{31}$$

Their interpretation is much simpler once expressed in terms of the scattering angle variables directly.

Let us introduce $z=\cos\theta$ and $z_i=\cos\theta_i$, where in the center-of-mass frame and notation of Fig. 3, $\theta$ is the scattering angle between $p_2$ and $p_3$, $\theta_1$ is the one between $p_2$ and $\ell$, and finally $\theta_2$ is the angle between $\ell$ and $p_3$. On the original integration contour, they satisfy the triangle inequalities: $\theta<\theta_1+\theta_2<2\pi-\theta$ and its two permutations, and are valued in $(0,\pi)$. After analytic continuation, neither variable needs to be in $(0,\pi)$, or even real. In terms of these variables, we have

$$P(z;z_1,z_2) = -\tfrac{1}{4}e^{-2i\theta}\left(e^{i\theta}-e^{i(\theta_1+\theta_2)}\right)\left(e^{i\theta}-e^{i(\theta_1-\theta_2)}\right)\left(e^{i\theta}-e^{-i(\theta_1-\theta_2)}\right)\left(e^{i\theta}-e^{-i(\theta_1+\theta_2)}\right).\tag{32}$$

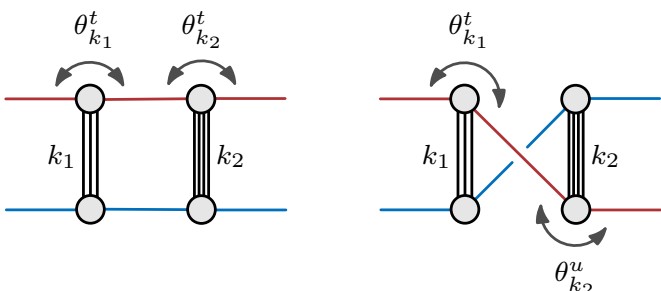

Figure 4: Two classes of singularities of the right-hand side of (1), obtained by gluing normal-threshold singularities of $T_1$ and $T_2$ with the specific values (36) respectively.

This immediately tells us that singularities at $\theta_i = \theta_i^*$ produce ones at $\theta = \theta^*$ with

$$\theta^* = \theta_1^* \pm \theta_2^*, \tag{33}$$

understood modulo $2\pi$. The solutions with $\theta^* \leftrightarrow -\theta^*$ are identified because they lead to the same $z = \cos\theta^*$.

Let us now look at concrete examples. The simplest singularities of $\theta_i^*$ are those responsible to $t_i$- and $u_i$-channel exchanges of the individual $T_i$. With a slight abuse of the notation, let us introduce

$$\theta_{k_i}^{t/u}(s) = \arccos\left[\pm\left(1 + \frac{2k_i^2 m^2}{s - 4m^2}\right)\right], \tag{34}$$

where the subscript $k_i$ refers to the number of particles exchanged and the superscript distinguishes between the channels: the $t_1$-channel exchange corresponds to $\theta_1^* = \theta_{k_1}^t(s)$ with a plus sign, $u_1$-channel to $\theta_1^* = \theta_{k_1}^u(s)$ with a minus sign, and likewise for $\theta_2^*$. Notice that $\theta_{k_i}^t(s) = \theta_{k_i}^u(s) + \pi$. Combining two such singularities gives rise to those at $\theta = \theta_{k_1,k_2}^*(s)$ with

$$\theta_{k_1,k_2}^*(s) = \theta_{k_1}^{t/u}(s) \pm \theta_{k_2}^{t/u}(s). \tag{35}$$

There are total three choices of signs, but only four possibilities are independent because

$$\theta_{k_1}^t(s) \pm \theta_{k_2}^t(s) = \theta_{k_1}^u(s) \pm \theta_{k_2}^u(s), \qquad \theta_{k_1}^t(s) \pm \theta_{k_2}^u(s) = \theta_{k_1}^u(s) \pm \theta_{k_2}^t(s), \tag{36}$$

modulo $2\pi$. We suppress the dependence on channels from the left-hand side of (35) to avoid clutter.

These singularities are illustrated in Fig. 4 and correspond to configurations in which all the $k_1 + k_2 + 2$ on-shell momenta are arranged in a planar and non-planar box diagram, for the two possibilities in (36) respectively, with either sign. The second option has a relative "twist" of the intermediate particles as a consequence of gluing $T_1$ and $T_2$ in mismatching channels.

Singularities of this type are known in perturbation theory as *anomalous thresholds* or *Landau singularities* [37–39], see [36, Secs. 3-4] for an introduction. Note that solving the above problem essentially rephrased Landau analysis in a non-perturbative setting: the $\delta^+$ function in (13) placed two particles on-shell, while requiring $\theta_i^* = \theta_{k_i}^{t/u}(s)$ imposed the on-shell constraints on the remaining $k_1 + k_2$ particles. Finally, the condition (30) is the same as the Landau loop equation requiring that the loop momentum $\ell$ lies in the space spanned by the external momenta. Together, these are the conditions imposing that the configurations in Fig. 4 can be interpreted as classical scattering processes with all the particle momenta localized to specific values. The power of the above derivation is that it gave a necessary and sufficient condition for a singularity, in contrast with the Feynman-diagram based analysis, which only gives a necessary condition.

We emphasize that here we made a particular choice of looking at the singularities with $\theta_i^*$ taking the specific form given by (34), but one can obtain much more complicated singularities in this way. In other words, those where the $T_1$ and $T_2$ blobs on the right-hand side of Fig. 3 can have more on-shell legs each. Not all of these singularities will appear on the second sheet. The most obvious singularity we do not consider are those obtained by gluing pseudo-normal thresholds, $\theta_i^* = 0, \pi$, or the triangle anomalous thresholds that we will come back to in Sec. 4.

The simplicity of the result (35) stems from the fact we used the scattering angles as our variables. To emphasize this point, let us express the same equation in terms of the Mandelstam invariants $(s, t)$. Plugging in $\cos\theta_{k_1,k_2}^*(s) = 1 + \frac{2t}{s-4m^2}$ above, we find that the singularities lie on the cubic surface:

$$4k_1^4 m^6 + 4k_2^4 m^6 - 8k_1^2 k_2^2 m^6 - k_1^4 m^4 s - k_2^4 m^4 s + 2k_1^2 k_2^2 m^4 s - 8k_1^2 m^4 t$$
$$+ 4k_1^2 k_2^2 m^4 t - 8k_2^2 m^4 t + 2k_1^2 m^2 st + 2k_2^2 m^2 st + 4m^2 t^2 - st^2 = 0 \qquad (37)$$

in the planar case. This equation contains solutions with both $\pm$ signs simultaneously. The planar case is obtained by replacing $t \to u = 4m^2 - s - t$ above. For low values of $k_1$ and $k_2$, we checked that they agree with the expressions obtained using elimination theory techniques for Landau equations [48].

## 3.2 Ladder-type Landau singularities

So far, we have only seen how to find singularities of the right-hand side of (1) in $z$ whenever those in $z_1$ and $z_2$ are already known. We will now make use of this analysis to consistently determine which singularities exist on the second sheet. The strategy will be to recursively leverage the fact that a singularity of the right-hand side of (1) implies that at least one term on the left-hand side has to be singular too.

On the first sheet, the only singularities of $T_1(s, z)$ are the one-particle poles and the multiparticle normal thresholds in the $s$ and $t, u$-channels, located at

$$s = k^2 m^2, \qquad z = \cos\theta_k^{t,u}(s), \qquad (38)$$

respectively for any $k = 1, 2, 3, \ldots$. As reviewed in Sec. 2.4, $T_2(s, z)$ has the same set of singularities, but can have more. For example, using the results of the previous subsection, we know that the right-hand side of the unitarity equation (1) can be singular at $z = \cos\theta_{k_1,k_2}^*(s)$. But none of these are singularities of $T_1(s, z)$ on the left-hand side. Therefore, for the equality to hold, $T_2(s, z)$ has to be singular in those places.

From this point on, we can proceed recursively. Since we have just found that $T_2(s, z_2)$ on the right-hand side of (1) is singular for any $\theta_2 = \theta_{k_2,k_3}^*(s)$ with positive integers $k_2$ and $k_3$, we have a new singularity of the right-hand side at $\theta_{k_1}^{t,u}(s) \pm \theta_{k_2,k_3}^*(s)$. This in general cannot be a singularity of $T_1(s, z)$ and so it has to correspond to that of $T_2(s, z)$. Repeating this argument over and over, we find that $T_2(s, z)$ has singularities at $\theta = \theta_{k_1,k_2,\ldots,k_n}^*(s)$ with

$$\theta_{k_1,k_2,\ldots,k_n}^*(s) = \sum_{i=1}^{n} \pm\theta_{k_i}^{t,u}(s), \qquad (39)$$

modulo $2\pi$, for any $n \geqslant 1$ and positive integers $k_i$ and the expression for $\theta_{k_i}^{t,u}(s)$ was given in (34). We suppress the $\pm$- and channel-dependence from the notation on the left-hand side. There are only two independent ways of sprinkling the $t/u$ assignments: either the number of $u$'s is even or odd, on top of the $2^{n-1}$ choices of signs. The even case was illustrated in Fig. 2, which we will refer to as the planar case. Singularities of this type have been found in perturbation theory and non-perturbatively using partial-wave analysis, see, e.g., [28,49,50].

A systematic way of utilizing similar rules has been described in [21] to identify $\mathbb{Z}_2$-symmetric Landau curves in the multi-particle region $16m^2 \leqslant s, t, < 36m^2$ on the physical sheet.

Note that we could have arrived at the same result if we simply analyzed the singularities of (26) coming from sewing together $t$- and $u$-channel singularities of $T_1$'s. However, we wanted to make sure that convergence of (26) does not affect the result and hence used the route above.

Once again, compactness of the result (39) emphasizes how the choice of the right variables can simplify the description of singularities. For example, let us look at the planar case with all $k_i = k$ and all plus signs. Expressed in terms of the Mandelstam invariants $(s, t)$ we have

$$1 + \frac{2t}{s - 4m^2} = \mathsf{T}_n\left(1 + \frac{2k^2 m^2}{s - 4m^2}\right), \tag{40}$$

where $\mathsf{T}_n$ are the Chebyshev polynomials of the first kind. The non-planar ladder case is obtained by relabeling $t \to u$. The resulting Landau discriminant is given by a degree-$n$ curve in $s$ and $t$. For instance, the case of the quadruple ladder ($n = 4$) gives

$$-64k^8 m^8 + 512k^6 m^8 - 128k^6 m^6 s - 1280k^4 m^8 + 640k^4 m^6 s$$
$$-80k^4 m^4 s^2 + 1024k^2 m^8 - 768k^2 m^6 s + 192k^2 m^4 s^2$$
$$-16k^2 m^2 s^3 - 64m^6 t + 48m^4 st - 12m^2 s^2 t + s^3 t = 0. \tag{41}$$

We verified the case $k = 1$ by solving Landau equations with numerical algebraic geometry software [48, Sec. 3.2]. For arbitrary $n$ and $k$, we find that the corresponding Schwinger proper times of all the $kn$ rung edges are equal to

$$\alpha_i = \frac{s - 4m^2}{2km^2}, \tag{42}$$

and those of the $2(n-1)$ side rail edges equal to $\alpha_j = 1$ (recall that Schwinger parameters are defined only up to an overall scale). Below, we will find that the numerator of (42) is always negative on the solutions of (40) with real scattering angles, meaning that all the $\alpha_i$'s of the rungs are negative. This is another way of seeing why such anomalous thresholds cannot lie on the first sheet, which requires $\alpha_i \geqslant 0$.

## 3.3 Natural boundaries

After identifying positions of the relevant singularities, let us now demonstrate that they densely accumulate on a half-line in the $s$-plane. In fact, to demonstrate the existence of a natural boundary, it will be sufficient to consider only a (measure zero) subset of the ladder-type singularities (39) with all $k_i$'s equal to $k$. The case $k = 1$ corresponds to ladders, relevant to any theory with cubic interactions. If a theory has a $\mathbb{Z}_2$ symmetry, $k = 2$ is the first contributing class of singularities. Without loss of generality, we work with the planar case and take all $\pm$ to be plus signs.

### 3.3.1 Fixed scattering angle

We first consider the case of the fixed scattering angle $\theta$, where it is the simplest to see the formation of a natural boundary. The second sheet has a singularity of the aforementioned type at every $s$ satisfying $\theta = n\theta_k^t(s) + 2\pi l$ for a fixed angle $\theta$. Here, we introduced the integer winding number $l$ to remind ourselves that the above equation is understood modulo $2\pi$. Inverting this equation identifies positions of the singularities of $T_2(s, \cos\theta)$ to be at $s = s_{k,l,n}^*(\theta)$ with

$$s_{k,l,n}^*(\theta) = 2m^2\left(2 + \frac{k^2}{\cos\left(\frac{\theta - 2\pi l}{n}\right) - 1}\right), \tag{43}$$

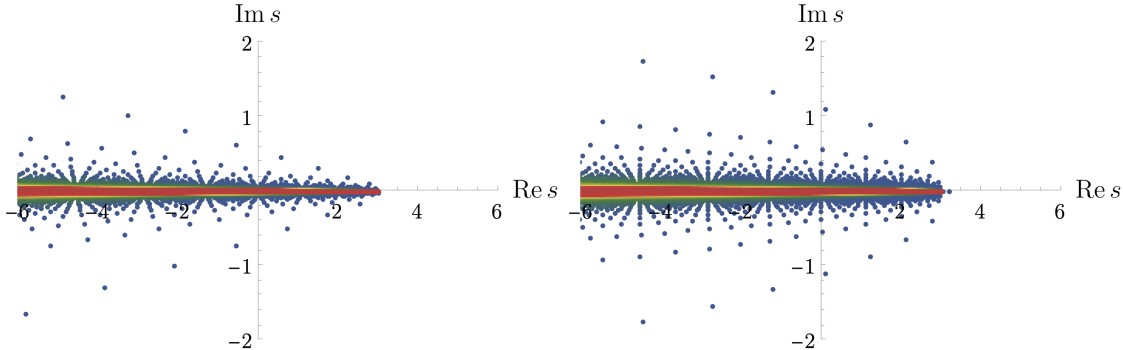

Figure 5: Subsets of ladder-type singularities with $k = 1$ and $m = 1$ up to $n \leqslant 500$ in the $s$-plane. In both cases, they accumulate on the half-line $s \leqslant 3$ (each plot features $\mathcal{O}(10^5)$ singularities). Higher values of $n$ are more red. Branch cuts are not plotted. **Left:** Fixed scattering angle $\theta = \frac{1+i}{2}$. All singularities collapse onto $s \leqslant 3$ as $\text{Im}\,\theta \to 0$. **Right:** Fixed momentum transfer $t = -1$. In the limit $t \to 0$ or $u \to 0$, all singularities lie on the axis $s \leqslant 3$.

where $l = 0, 1, \ldots, n-1$. Let us now consider the cosine in the limit as $n$ becomes very large compared to $\theta$:

$$\cos\left(\frac{\theta - 2\pi l}{n}\right) = \cos\left(\frac{2\pi l}{n}\right) + \frac{\theta}{n}\sin\left(\frac{2\pi l}{n}\right) + \mathcal{O}\left[\left(\frac{\theta}{n}\right)^2\right]. \tag{44}$$

Since the fractions $\frac{l}{n}$ densely cover the unit interval, the image of the cosine densely covers $[-1, 1]$. Hence, regardless of the scattering angle $\theta$, for any fixed $k$, a dense accumulation of singularities develops on the half-line

$$s \leqslant (4 - k^2)m^2, \tag{45}$$

on the real axis in the $s$-plane.

Let us notice that for any physical (real) $\theta$, all $s^*_{k,l,n}$ are real and hence all the singularities lie on the half-line (45). As shown above, the natural boundary still exists after complexifying the scattering angle, but the singularities are more spread out around it and approach the real axis as $n \to \infty$, see Fig. 5 (left).

For example, the natural boundary illustrated in Fig. 1 (right) extends throughout $s \leqslant 3m^2$ in theories with three-particle interactions, after a resummation of $k = 1$ ladders. For a $\mathbb{Z}_2$-symmetric theory, we would have to sum over $k = 2$ ladders and the natural boundary would exist for $s \leqslant 0$.

### 3.3.2 Fixed momentum transfer

The analysis at fixed momentum transfer $t$ follows similar steps. Let us first consider the case of the forward limit, $t = 0$ (or backward, $u = 0$). Since it is the same as the case of the fixed angle $\theta = 0$ (or $\theta = \pi$), we immediately conclude that a natural boundary develops for (45) in those cases. Notice that the forward/backward limit is even more singular, because positions of singularities depend solely on the ratio $\frac{l}{n}$ in (43). Hence, two planar ladders with $n_1$ and $n_2$ rungs will have coinciding singularities unless $(n_1, n_2)$ are coprime. It is not clear if this additional accumulation could give rise to moving of singularities similar to the 1PI resummation.

Away from forward limits, the singularities are more spread out in the complex $s$-plane at fixed $t$ and it is more difficult to write down a closed-form expression for the positions of

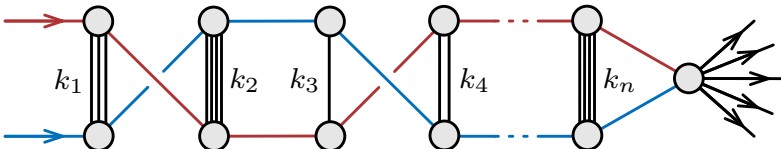

Figure 6: The simplest class of singularities expected to exist for a $2 \to$ any process.

singularities at arbitrary $n$. One way to determine them is to use the equation (43), except now evaluated at the $s$- and $t$-dependent scattering angle $\cos\theta(s,t) = 1 + \frac{2t}{s-4m^2}$. At large $n$, we thus obtain a good approximation by using (44) and plugging in the approximate value $\cos\theta(s^*_{k,l,n}, t) \approx 1 + \frac{t}{k^2 m^4}\left(\cos\left(\frac{2\pi l}{n}\right) - 1\right)$, giving

$$s^*_{k,l,n}(t) \approx 2m^2\left(2 + \frac{k^2}{\cos\left(\frac{2\pi l}{n}\right) + \frac{1}{n}\arccos\left[1 + \frac{t}{k^2 m^4}\left(\cos\left(\frac{2\pi l}{n}\right) - 1\right)\right]\sin\left(\frac{2\pi l}{n}\right) - 1}\right), \quad (46)$$

where $l = 0, 1, \ldots, n-1$ as before. Since the arccosine drops out in the $n \to \infty$ limit, we obtain the same leading behavior as in the fixed-angle case and singularities accumulate densely on $s \leqslant (4-k^2)m^2$ independently of the value of $t$. For any finite $n$, singularities are spread out around this half-line, see Fig. 5 (right).

## 4 Discussion

In this work we demonstrated that $2 \to 2$ scattering amplitudes in gapped theories can have a natural boundary on the second sheet of the lightest-threshold branch cut. This realization opens up a number of questions and future directions, some of which are outlined below.

**Higher multiplicity.** The most obvious question is whether natural boundaries can be generic analytic features of scattering amplitudes at higher multiplicity. Let us argue that this is indeed the case.

Extending the analysis of Sec. 3, one finds other classes of singularities that develop on the second sheet. Arguably, the next-to-simplest family is comprised of the triangle-ladders illustrated in Fig. 6. Repeating the analysis leading to (39), one finds that positions of triangle-ladder singularities with $n$ rungs are given by sending $t = 0$ or $u = 0$ on the rectangular ones, i.e. they are determined by

$$\theta^*_{k_1, k_2, \ldots, k_n}(s) = 0 \ \text{ or } \ \pi, \quad (47)$$

which depends only on $s$. Explicitly, for $k_i = k$ and all plus signs, we have a singularity for every

$$s^*_{k,l,n} = 2m^2\left(2 + \frac{k^2}{\cos\left(\frac{(0 \text{ or } \pi) + 2\pi l}{n}\right) - 1}\right), \quad (48)$$

where $l = 0, 1, \ldots, n-1$. As before, these accumulate to form a natural boundary on $s \leqslant (4-k^2)m^2$.

Similarly to the case discussed in Sec. 3.3.2, for every prime number $p$, triangle-ladders with $n = p, 2p, 3p, \ldots$ rungs will share common singular points. They might potentially resum in a way similar to the 1PI resummation of simple poles, thus moving the position of the singularity. We leave an investigation of this question until future work.

The triangle-ladder singularities also exist for $2 \to$ any and any $\to 2$ scattering processes, because one can add external particles without modifying the position of the singularity in

the center-of-mass energy variable $s$, see Fig. 6. This fact suggests the existence of natural boundaries on the second sheet of the lightest-threshold branch cuts for processes of arbitrary multiplicity. Certainly, this is only a tip of the iceberg since much more complicated singularities can arise as well.

**Explicit checks.**    Another natural question is whether the results of this work can be validated with explicit computations in perturbation theory. As emphasized above, a natural boundary cannot be seen at any finite loop level, but some evidence can be gathered by demonstrating that the individual singularities comprising the would-be natural boundary do indeed exist in a given process. To start answering this point, one would wish to at least evaluate the class of diagrams creating the boundary, i.e. the ladders or triangle-ladders with $n$ rungs and a single mass $m$. At least currently, such a computation seems far out of reach, because the type of functions encountered in such a calculation would have to be *at least* as complicated as periods of Calabi–Yau $(n-1)$-folds [51]. Since positions of singularities become much simpler to express in terms of the $(s, \theta)$ variables, as opposed to $(s, t)$, one would expect that finding analytic expressions, or perhaps differential equations, for such Feynman integrals would benefit from using the same variables.

Another direction is to investigate a possibility of natural boundaries and the analogue of the ladder-type singularities on the second sheet also in conformal field theories, perhaps in relation to [52, 53].

**Sufficiency.**    Let us further scrutinize the results by asking what would it take for the natural boundary to disappear in a given theory. Recall that presence of singularities of $T_2$ is rather strongly constrained by unitarity. Hence if $T_2(s, z)$ were to be non-singular for some fixed $s$ predicted by (39), the corresponding singularities would have to cancel among each other on the right-hand side of (26). For such a situation to take place, the cancellation would have to happen either after summing over different species (of the same mass) running through the cut, or perhaps between different cuts that are singular at the same kinematic point. For instance, an $n$-rung ladder with all plus signs in (39) is singular at the same $\theta$ as an $(n+2)$-rung ladder with a single minus sign. Investigation of such cancellation effects would likely have to involve working with a specific theory.

**Massless theories.**    Several obstructions exist for extending our results to quantum field theories containing massless particles. First, the definition of the second sheet required us to be able to isolate the elastic region (above the lightest-threshold branch cut, but below the next-to-lightest) and hence now breaks down. Secondly, even without knowing the sheet structure, one might wonder where do the ladder-type Landau singularities lie in the $s$-plane in the massless case. Using (39) shows that they always correspond to $t = 0$ or $u = 0$, on top of the $s = 0$ branch points evident from the unitarity equation. Hence, for an infinite number of *distinct* singularities to exist, one needs at least one massive particle in the spectrum. Finally, despite progress [8, 48, 54–56], Landau analysis has never been formulated precisely enough in theories with IR divergences and hence would have to be developed first.

# Acknowledgments

S.M. thanks Nima Arkani-Hamed, Pinaki Banerjee, Ruth Britto, Simon Caron-Huot, Miguel Correia, Lorenz Eberhardt, Carolina Figueiredo, Alfredo Guevara, Hofie Hannesdottir, Aaron Hillman, Yuqi Li, Daniel Longenecker, Dalimil Mazac, Matteo Parisi, and Wayne Zhao for useful discussions. S.M. gratefully acknowledges funding provided by the Sivian Fund. This material

is based upon work supported by the U.S. Department of Energy, Office of Science, Office of High Energy Physics under Award Number DE-SC0009988.

# A   Analyticity at fixed momentum transfer

In this appendix, we review aspects of physical-sheet analyticity in the complex $s$-plane at fixed momentum transfer squared $t < 0$. This case turns out to be much more complicated than analyticity at fixed scattering angle employed in the main part of the paper and briefly reviewed in App. B.

As the starting point, we decompose the connected part of the $2 \to 2$ S-matrix elements, which can in principle depend on the quantum numbers $\lambda_i$ of each particle, schematically as a finite sum

$$T^{\lambda_1, \lambda_2; \lambda_3, \lambda_4}(p_1, p_2; p_3, p_4) = \sum_i a_i^{\lambda_1, \lambda_2; \lambda_3, \lambda_4} \, T_i^s(s, t), \qquad (49)$$

where $a_i$ form a basis of representation-dependent tensor structures (for example, contractions of momenta $p_i$, polarization vectors $\epsilon_i$, and colors in gauge theory) and $T_i^s$ are functions that depend purely on the Mandelstam invariants $s$ and $t$. The subscript $s$ is here to emphasize that $T_i^s$ computes the $s$-channel scattering, $12 \to 34$. Historically, (49) has been called the *Joos expansion* and in the modern literature it is known as *tensor decomposition*, see, e.g., [57–59].

The problem of studying analytic properties of $2 \to 2$ scattering amplitudes therefore decomposes into understanding singularities of $a_i$ and $T_i^s$ separately, after complexification. The former has been studied for various choices of external states in [60–66] and more recently in [67–69], among others. The analytic properties of the latter are much less understood and we focus on them now.

The problem is often framed in the context of crossing symmetry, which is trying to show that $T_i^s(s, t)$ and its counterparts in the other channels, $T_i^t(s, t)$ and $T_i^u(s, t)$, are analytic continuations of the same function. More concretely, it supposes that for every $i$, there exists a single function $T_i(s, t)$ analytic in some region of the $(s, t)$-space connecting the kinematic channels as follows:

$$\lim_{\varepsilon \to 0^+} T_i(s + i\varepsilon, t) = T_i^s(s, t), \qquad \lim_{\varepsilon \to 0^+} T_i(s, t + i\varepsilon) = T_i^t(s, t), \qquad (50)$$

$$\lim_{\varepsilon \to 0^+} T_i(s - i\varepsilon, t) = \lim_{\varepsilon \to 0^+} T_i(s, t - i\varepsilon) = T_i^u(s, t). \qquad (51)$$

In each case, the values of $(s, t)$ are taken in the corresponding physical region, carved out by the Gram condition $\det\left[p_i \cdot p_j\right]_{i,j=1,2,3} > 0$ and $s > 0$, $t > 0$, or $u > 0$ respectively. It is known that (50-51) might be violated if the external states are unstable [36], so it is important to assume stability of every state in the theory at this stage. While not entirely obvious, one can show that in those cases the above $i\varepsilon$ agrees with the Feynman $i\varepsilon$ prescription for propagators.

## A.1   Bros–Epstein–Glaser analyticity

Let us fix $t < 0$ and consider analyticity in the $s$-plane. Unitarity dictates presence of branch cuts, which by convention are chosen be $s < \sum_{i=1}^4 p_i^2 - 4m^2 - t$ (translating to $u > 4m^2$) and $s > 4m^2$ coming from two-particle normal thresholds in the $u$- and $s$-channels respectively, where $m$ is the mass of the lightest exchanged state. Whenever $|t|$ is sufficiently small, the two kinds of branch cuts do not overlap and the region between them is called the *Euclidean region*. If such a region exists, one can prove analyticity in the whole $s$-plane minus the normal-threshold cuts [70]. One can also see this fact to arbitrary loop order in perturbation theory using a derivation analogous to that given in App. B.

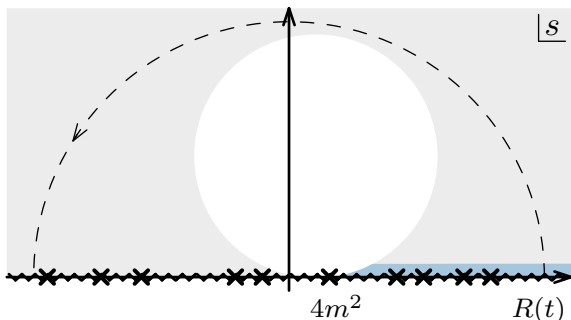

Figure 7: Known domain of analyticity (shaded) of $2 \to 2$ scattering amplitudes in the upper half-plane of $s$ at fixed $t < 0$ with no Euclidean region for theories with a mass gap $m > 0$. The only possible singularities in the upper half-plane can exist within the unshaded area. There exists a sufficiently large $R(t)$ such that the amplitude is analytic everywhere in $|s| > R(t)$ with $\mathrm{Im}\, s > 0$ (outside of the dashed arc). The $s$-channel physical region, indicated in blue, is contained in the region of analyticity. The dashed arc is the path of analytic continuation explained in the text.

The trouble starts when the Euclidean region ceases to exist, e.g., when $t \leqslant \sum_{i=1}^{4} p_i^2 - 8m^2$. In this case, under certain assumptions, Bros, Epstein, and Glaser [71,72] proved that for every $t < 0$, there exists a sufficiently large radius $R(t)$ such that the amplitude is analytic in the upper half-plane in $s$ with

$$|s| > R(t), \tag{52}$$

see Fig. 7. There is presently no concrete estimate for $R(t)$ in terms of the mass gap $m$, momentum transfer $t$, or other scales in the problem, so in principle $R(t)$ may be larger than the Planck scale. It is known that asymptotically as $t \to -\infty$, $R(t)$ grows at least as $|t|^3$ [73, Sec. 5.3]. Since the proofs of [71,72] break down at the very first step once a massless particle is present in the spectrum, one could expect that $R(t) \to \infty$ in that limit. It is generally believed that the region in Fig. 7 is far from optimal, but improving it seems highly technical and remains an open problem. In Sec. A.2, we give a toy model illustrating how this region was obtained in the first place.

Analytic continuation along the dashed curve in Fig. 7 connects the $s$-channel physical region to the $u$-channel but approached with the wrong sign of the $i\varepsilon$, cf. (51). Nevertheless, at this stage one can perform a similar analytic continuation with flipped $s \leftrightarrow t$ (large $|t|$ with $\mathrm{Im}\, t > 0$ and $s < 0$ fixed) to arrive at the $t$-channel physical region approached from the correct side. This achieves crossing symmetry in the scalar case. More work is required for S-matrices involving spinning particles, because one needs to simultaneously continue the $a_i$'s in (49). Some explicit results include [62, 65], but the spinning case does not appear to be entirely settled.[2]

Other than the aforementioned mass gap and stability conditions, Bros–Epstein–Glaser analyticity assumes unitarity, UV finiteness, existence of local operators, and their space-like commutativity. Recent work on the flat-space limit of AdS/CFT correlators hints that the existence of local operators in the bulk might not be needed to establish analyticity of IR-safe S-matrices [77–79]. Results on isolating the constraints coming from causality without employing unitarity are given in [80]. Some work on weakening the space-like commutativity condition includes [81]. Certain intuition for the region of analyticity can be obtained by studying commutativity of operators in the Regge limit [82]. It is generally difficult to reconcile such arguments with on-shell conditions [83, App. A]. Only limited results are known in

---

[2]We believe that the modified crossing rules observed in Chern–Simons theories [74–76] have a similar origin.

perturbation theory; in the planar limit one can show analyticity in the whole upper half-plane to all loop orders, even in gapless theories [83].

For a more thorough review of analytic properties, including those in two variables $(s,t)$ simultaneously, we refer to [73].

## A.2 Toy model for analytic completion

In this section we sketch the type of manipulations needed to establish the domain of analyticity in Fig. 7.

The starting point is the convergence analysis of Fourier transforms of correlations functions of retarded commutators from coordinate to momentum space. Its gist is that after imposing microcausality, i.e. vanishing of commutators at space-like separations, one encounters factors of the type $e^{ip_j \cdot x_j}$ (in mostly-minus conventions), where $x_j$ is only allowed to be time-like. It means that for the exponential suppression of the integrand one needs $\text{Im}\, p_j \cdot x_j > 0$, restricting $\text{Im}\, p_j$ to be time-like or null. This, in turn, is not compatible with on-shell conditions in a physical theory, which in particular impose that masses-squared $p_j^2 = M_j^2$ are non-negative. To be more concrete, $\text{Im}\, M_j^2 = 2\,\text{Re}\, p_j \cdot \text{Im}\, p_j = 0$ implies that $\text{Re}\, p_j$ has to be space-like, which leads to $\text{Re}\, M_j^2 = \text{Re}\, p_j^2 - \text{Im}\, p_j^2 < 0$, giving a contradiction. This toy model already illustrates the ubiquitous tension between analyticity and on-shellness intrinsic to the LSZ formulation.

One is therefore forced to work with off-shell Green's functions in momentum space, depending on the parameters $(s, t, M_1^2, M_2^2, M_3^2, M_4^2) \in \mathbb{C}^6$, as opposed to on-shell scattering amplitudes defined purely in the space of the Mandelstam invariants $(s,t) \in \mathbb{C}^2$. A more careful analysis of the above convergence properties guarantees the so-called *primitive domain* of analyticity in $\mathbb{C}^6$ that does not intersect the mass shell: for each proper subset of particles $S$ with the total momentum $p_S$, it requires that either $\text{Im}\, p_S$ is time-like or null, or $\text{Im}\, p_S = 0$ and $p_S^2 < M^2$ for any production threshold $M^2$ [84–88]. Of course, this procedure also comes with the usual catalog of assumption making LSZ formalism well-defined, including existence of local operators and a finite mass gap in the spectrum, UV finiteness, stability, etc.

Some intuition for the connection between causality and analyticity can be gained from studying $(0+1)$-dimensional scattering problems, see, e.g., [89] and [90, App. D]. The simplest version is to consider a signal $f(t)$ as a function of time $t$ and its Fourier transform to the energy space:

$$\tilde{f}(E) = \int_{-\infty}^{\infty} dt\, e^{iEt} f(t). \tag{53}$$

If we assume that the signal was a response to an event happening at $t = 0$, the integrand has support only for $t \geqslant 0$, encoding a notion of causality. Convergence at infinity requires $\text{Im}(Et) > 0$, which guarantees analyticity of $\tilde{f}(E)$ in the upper half-plane of $E$. As emphasized above, such arguments cannot be easily generalized to $2 \to 2$ scattering in relativistic theories precisely because of the higher-dimensional nature of the problem: it is the fact that multiple overlapping scattering channels (one for each $S$) are simultaneously allowed. Another obstruction is mapping to the Mandelstam invariants which are quadratic in energies and momenta.

At this stage, the strategy is to turn to a rather technical discussion of theorems in analytic completion of domains of holomorphy. It is a feature of analytic functions in several complex variables without any counterparts in one variable.[3] Given some domain $D$, one can construct a larger domain $\mathcal{H}(D)$ called the *envelope of holomorphy* of $D$, such that *any* holomorphic function in $D$ admits a unique analytic continuation to $\mathcal{H}(D)$. There is no general practical way

---

[3]The theory of analytic functions in several complex variables is virtually never taught in physics courses. We recommend the textbooks [91–93] for elementary introductions and [94–96] for more in-depth material needed for applications to quantum field theory.

of finding $\mathcal{H}(D)$. Rather, there exist multiple "edge of the wedge"-style theorems that can be used depending on the application at hand, see, e.g., [73, Sec. 2.1] for a review. One of the simplest examples is the Bochner tube theorem [97]. Recall that a *tube* $T_B$ is a generalization of a strip to several variables and in $m$ dimensions can be written as

$$T_B = B + i\mathbb{R}^m, \tag{54}$$

where the base $B$ is some region in $\mathbb{R}^m$ and the imaginary parts remain unconstrained. The tube theorem says that the envelope of holomorphy of such a tube is

$$\mathcal{H}(T_B) = T_{\mathrm{ch}(B)}, \tag{55}$$

obtained by taking the convex hull $\mathrm{ch}(B)$ of the base $B$. This result can be powerful when $B$ is very non-convex to start with, since it buys us a lot of analyticity. A number of problems in analytic completion can be solved with variations of the tube theorem.

The goal of the following paragraphs is to give the reader an idea of how analytic completion theorems of the above type can be used to derive the region of analyticity shown in Fig. 7. To this end, we consider a toy model which illustrates all the essential features in a simplified setting. A different toy model was presented in [98]. For the full proof, see [71, 72].

It will be sufficient to work in a two-dimensional subspace of the off-shell kinematic space $\mathbb{C}^6$ parameterized by $(\zeta, s) \in \mathbb{C}^2$, where $\zeta = \sum_{i=1}^4 p_i^2$ measures the off-shell deformation while satisfying momentum conservation $\zeta = s + t + u$. The on-shell point is thus located at $\zeta = \zeta_*$ with

$$\zeta_* = \sum_{i=1}^4 M_i^2. \tag{56}$$

Let us take it for granted that one can construct two domains in the $(\zeta, s)$-space where analyticity can be proven, $D_1$ and $D_2$, illustrated in Fig. 8.

Both domains can be written as product spaces. The first one, $D_1$ is a neighborhood of some off-shell point $\zeta_0 < 0$ (say, a circle with a small radius $r_1$) times the upper half-plane in $s$, see Fig. 8 (top). This is a toy model for the aforementioned primitive region of analyticity. On the other hand, $D_2$ is a larger domain in $\zeta$ containing $\zeta_0$ and the on-shell point $\zeta_*$ (say, a circle with a large radius $r_2$) times the infinitesimal neighborhoods of the physical regions $s < s_1$ and $s > s_2$, see Fig. 8 (bottom). This region guarantees that we can define the on-shell S-matrix in the first place, by approaching the physical regions from the upper half-plane. The question becomes whether a function defined in $D_1 \cup D_2$ (an off-shell Green's function) can be simultaneously continued on-shell and analytic in some finite region in the upper half-plane of $s$.

The domain $D_1 \cup D_2$ is not yet a tube, but can be easily converted to one with the change of variables

$$\zeta' = \log(\zeta - \zeta_0), \qquad s' = i \log\left(\frac{s - s_2}{s - s_1}\right). \tag{57}$$

Under this map, the imaginary parts $(\mathrm{Im}\,\zeta', \mathrm{Im}\,s')$ of the above domain are unconstrained, but the real parts $(\mathrm{Re}\,\zeta', \mathrm{Re}\,s')$ belong to some region $B = B_1 \cup B_2$, which is the base $B$ of the tube $T_B$, see Fig. 9 (left). In order to perform analytic completion, all we have to do is to determine the convex hull of $B$ illustrated in Fig. 9 (right). The result is the interpolating family of direct products:

$$\mathrm{ch}(B) = \bigcup_{0 \leqslant \lambda \leqslant 1} \left\{\mathrm{Re}\,\zeta' < \log\left(r_1^\lambda r_2^{1-\lambda}\right)\right\} \times \left\{0 < \mathrm{Re}\,s' < \lambda\pi\right\}. \tag{58}$$

Translating back to the original variables $(\zeta, s)$, the envelope of holomorphy is given by

$$\mathcal{H}(T_B) = \bigcup_{0 \leqslant \lambda \leqslant 1} \left\{0 \leqslant |\zeta - \zeta_0| < r_1^\lambda r_2^{1-\lambda}\right\} \times \left\{0 < \arg\left(\frac{s - s_2}{s - s_1}\right) < \lambda\pi\right\}, \tag{59}$$

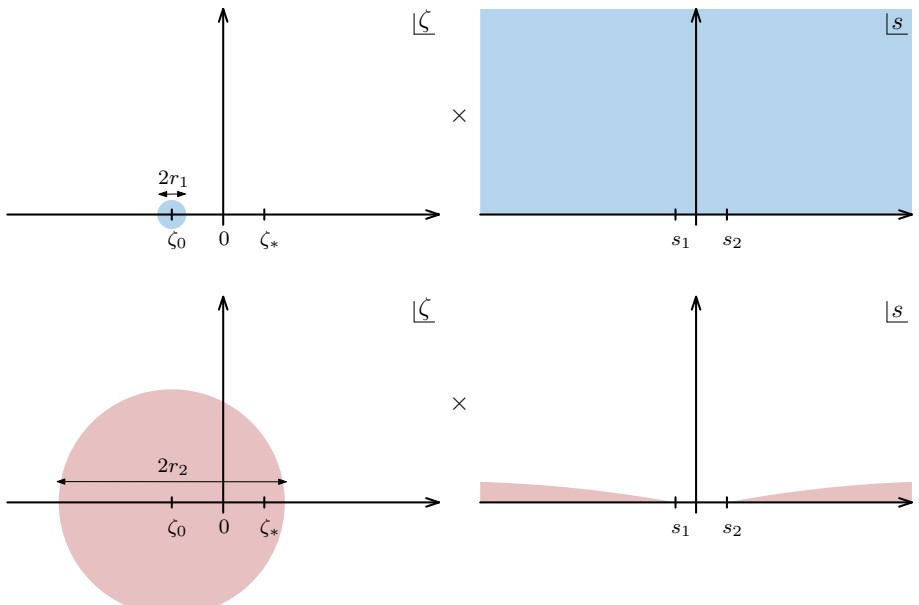

Figure 8: The domains of analyticity $D_1$ (top) and $D_2$ (bottom) in the $\zeta$- and $s$-planes (shaded). **Top:** $D_1$ is the direct product of a small circle around $\zeta_0 < 0$ with radius $r_1$ times the upper half-plane in $s$. **Bottom:** $D_2$ is the direct product of the circle around $\zeta_0$ with radius $r_2 > r_1$ large enough to contain the on-shell point $\zeta_* = \sum_{i=1}^{4} M_i^2$, and the infinitesimal upper half-plane neighborhoods of the physical regions $s < s_1$ ($u$-channel) and $s > s_2$ ($s$-channel) for some $s_1 < s_2$.

which for every $\lambda$ is a circle in the $\zeta$-plane times a subset of the $s$ upper half-plane *outside* of a circle intersecting $s_1$ and $s_2$ with a certain radius determined by the constants $\zeta_0, r_1, r_2, s_1, s_2$. The latter is precisely the domain of the type presented in Fig. 7.

To be more concrete, let us put a bound on how large $|s|$ needs to be to guarantee analyticity (what was called $R(t)$ in Fig. 7). The smallest $\lambda$ still including the on-shell point $\zeta_*$ is

$$\lambda_* = \log\left(\frac{\zeta_* - \zeta_0}{r_2}\right)\bigg/\log\left(\frac{r_1}{r_2}\right), \tag{60}$$

which determines the lower bound on the size of the circle in the $s$-plane, after some straightforward algebra:

$$|s| > \max_{x>0}\sqrt{\frac{x^2 s_1^2 - 2x s_1 s_2 \cos(\lambda_* \pi) + s_2^2}{x^2 - 2x\cos(\lambda_* \pi) + 1}}. \tag{61}$$

One can solve the extremization problem to express the right-hand side in terms of elementary functions, but we do not do it here for conciseness.

The problem of finding the analyticity domain of $2 \to 2$ scattering amplitudes follows steps of similar flavor, but is much more involved technically [71, 72, 99] (see also [100, 101] for higher multiplicity). In particular, let us emphasize that finding the full envelope of holomorphy is an extremely difficult problem and so far we only know about a partial analytic completion. As a consequence, the region of analyticity in Fig. 7 is expected to be far from optimal. As a matter of fact, what has been shown is only that for any $t < 0$, there exists sufficiently large $R(t)$ such that the amplitude is analytic in $|s| > R(t)$ in the upper half-plane, but no concrete bound of the type (61) has been found in terms of the mass gap and other scales in the problem.

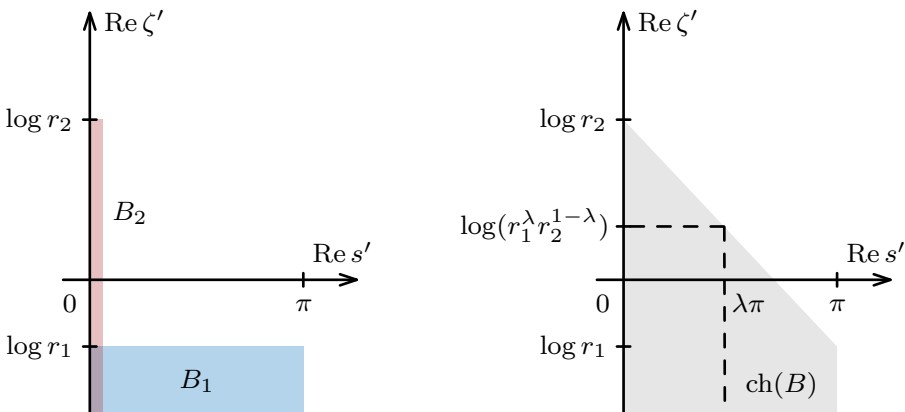

Figure 9: Application of the tube theorem. **Left:** The base $B = B_1 \cup B_2$ consists of two infinite rectangles corresponding to the real parts of $(\zeta', s')$. **Right:** The convex hull $\mathrm{ch}(B)$ can be described as a family of infinite rectangles (enclosed by dashed lines) parameterized by $\lambda \in [0, 1]$ that interpolate between $B_1$ and $B_2$.

## B  Analyticity at fixed scattering angle

In this appendix we prove that for any fixed real $z = \cos\theta$, the amplitude $T(s, z)$ is analytic in the $s$-plane minus the normal threshold branch cuts to all orders in perturbation theory. We use completely different techniques to those of App. A for variety. The proof is a special case of a more general result [102] after choosing four external states and a particular slicing through the kinematic space, and hence we will keep the discussion brief. To match the main text, we consider a theory with a single particle with mass $m > 0$.

Recall that analytic properties of any given Feynman integral are dictated by the action $\mathcal{V}$ depending on the kinematic invariants and the Schwinger parameters $\alpha_e$ of the internal edges $e$. The corresponding Feynman integral is analytic around a given point $(s, z)$ provided that $\mathcal{V} = \mathcal{V}(s, z)$ is non-zero for all non-negative values of Schwinger parameters. In the notation of [36, Sec. 4], it takes the form

$$\mathcal{V}(s, z) = s\mathcal{V}_s + \tfrac{1}{2}(s - 4m^2)\Big((z-1)\mathcal{V}_t - (z+1)\mathcal{V}_u\Big) + m^2\Big(\sum_{i=1}^{4}\mathcal{V}_i - \sum_{e=1}^{\mathrm{E}}\alpha_e\Big). \tag{62}$$

Different terms are determined by the topology of the Feynman diagram. The only thing that matters to us is that $\mathcal{V}(s, z)$ is linear in $s$ and remains real for real $s$.

Recall that the kinematic space has a Euclidean region containing points $(s_*, z)$ with $s_* \in (4m^2\frac{z-1}{z+1}, 4m^2)$ for any $z \in (-1, 1)$. It corresponds to the kinematics below all possible two-particle thresholds where the amplitude is real aside from possible poles, see Fig. 1 (left). Another way to characterize it is that

$$\mathcal{V}(s_*, z) < 0, \tag{63}$$

for any Feynman integral and all admissible values of Schwinger parameters. Let us pick any such $s_*$ as a reference point. Due to linearity of $\mathcal{V}(s, z)$, we have

$$\mathcal{V}(s, z) = \mathcal{V}(s_*, z) + (s - s_*)\partial_s\mathcal{V}(z). \tag{64}$$

For a singularity or a branch cut to exist, we need $\mathcal{V}(s, z) = 0$. Let us take $s$ in the upper or lower half-plane and assume that such a non-analyticity develops. In particular, it would require

$$\mathrm{Im}\,\mathcal{V}(s, z) = (\mathrm{Im}\,s)\,\partial_s\mathcal{V}(z) = 0. \tag{65}$$

Since we already assumed that $\mathrm{Im}\, s \neq 0$, the above equation forces $\partial_s \mathcal{V}(z) = 0$. But in such a situation

$$\mathrm{Re}\, \mathcal{V}(s,z)\big|_{\partial_s \mathcal{V}(z)=0} = \mathcal{V}(s_*,z) < 0\,, \tag{66}$$

by (63), which means that $\mathcal{V}(s,z) \neq 0$. Hence there are no singularities or branch cuts anywhere away from the real axis. Moreover, the amplitude is real on the interval on the real $s$-axis given by the Euclidean region and hence it is given by a unique analytic function in the complex $s$-plane minus the normal-threshold branch cuts.

To complete the analysis, one should also be able to demonstrate that approaching the $s$-channel branch cuts from the upper (as opposed to lower) half-plane recovers the original amplitude consistent with the Feynman $i\varepsilon$, i.e. that $\mathrm{Im}\, \mathcal{V}(s,z) > 0$ as $s$ approaches the branch cut. To see this, notice that the $s$-channel normal thresholds are always positioned at $s \geqslant 4m^2$, i.e. $s > s_*$. From (64), we see that $\mathcal{V}(s,z) = 0$, which is the condition for a branch cut, can only be attained for $s > s_*$ when $\partial_s \mathcal{V}(z) > 0$. Hence the requirement that $\mathrm{Im}\, \mathcal{V}(s,z) > 0$ implies $\mathrm{Im}\, s > 0$.

An entirely analogous derivation can be repeated for fixed-$t$ Feynman integrals and shows that they are analytic in the $s$-complex plane minus cuts as long as $t \in (-4m^2, 4m^2)$, such that the real $s$-axis intersects the Euclidean region. As reviewed in App. A.2, extending this argument to more general situations is much more difficult, and in particular remains an open problem for $m = 0$.

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
