# Peer review of "Natural Boundaries for Scattering Amplitudes"

_SciPost Physics, doi:SciPost Phys. 14, 101 (2023)_

## Round 1 · Referee Report · Jnanadeva Maharana (Referee 1) · 2022-12-19

Strengths

This article has important results for study of analyticity of scattering amplitude. Although this area is currently not popular, there important issues to be examined. The author has systematically studied several
problems.

Weaknesses

I really do not find serious weaknesses in the paper.

Report

The contents of the article is quite interesting and definitely this work deserves publication in your journal.

Requested changes

No major revision is required.

---

## Round 1 · Referee Report · Anonymous (Referee 2) · 2023-1-2

Strengths

1- The paper is well-written 2- The arguments are presented in detail and rigorously

Weaknesses

1- While the existence of a dense set of singularities seems rigorous enough, it is not clear to me that these singularities belong to the second sheet. If they don't then it could still be possible to analytically continue through the apparent barrier on the second sheet since some (most?) of these singular points actually belong to other sheets. 2- Another way to express the same thing is that there is an infinite number of choices of $\pm i \epsilon$ prescriptions for all the propagators, or an infinite number of choices of signs of energies in cut propagators. Each one of these choices corresponds to picking another sheet as the starting point of analytic continuation. Somehow the arguments in this paper would say that all these other sheets are not all connected by analytic continuation, but I believe they all exist and can be naturally associated to the S-matrix element under study. In other words, it seems to me that even if all these singularities are actually on the second sheet, in some sense the amplitude can be defined on other sheets, even though this will not be via an analytic continuation through the "boundary".

Report

Happy New Year!

The paper is well-written and draws attention to phenomena that arise in the non-pertubative amplitudes which are not apparent when studying the perturbative amplitudes. This is valuable, even if similar points have been made in older, mostly forgotten, literature. I recommend the paper for publication, but I would like the author to spend more time on the discussion of whether the singularities actually belong to the second sheet, following the discussion in Weaknesses above.

Requested changes

1- It would be interesting to see if there are some examples functions (perhaps in a single variable, to keep things simple) defined via integrals which have singularity barriers. The examples I know (and the ones that are presented in the paper) are defined by power series.

2- Below eq. 2.11 when discussing the normalization please mention that the normalization is fixed by unitarity. I believe that there should be some $(-2 \pi i)^2$ terms missing and possibly some $\frac 1 2$ from the integration over the phase space of identical particles. I don't want to demand a change here, but only to preempt some people's puzzlement at the equation.

3- Below eq. 2.25, please explain in a bit more detail the notion of Mandelstam analyticity.

4- Also below eq. 2.25, $\Im T(s, z)$ is only real-analytic, right?

5- Below eq. 3.1, missing continue'' in the phrase starting withAs we analytically...''.

6- Below eq. 3.6, in the definition of $\theta_1^*$ perhaps what was meant was $\theta_1^* = \theta_{k_1}^t(s)$?

7- In the last paragraph of page 15, ``rephrased Landau analysis in a non-perturbative setting'', but the Landau analysis does not have to be perturbative in the first place. Landau made this point in his original paper. Indeed one can consider Landau diagrams with vertices being non-perturbative S-matrix elements. These kind of considerations have been used in two-dimensional integrable theories.

8- Explain fig. A.1 in more detail. What is the white disk? Presumably $|s| = R(t)$ and $\Im s > 0$ is the dashed line...

9- At the bottom of page 24, Since the proofs of [] breaks down'', replace bybreak down''.

---

## Round 2 · Referee Report · Anonymous (Referee 2) · 2023-1-17

Report

The changes look good. The paper can be published as far as I'm concerned.

---

## Round 2 · Author Response

I would like to thank both referees for the time spent reviewing the paper and their reports.

First, I clarified two misunderstandings brought up by Referee 2. Second, I revised the paper to incorporate the comments/questions they raised and included an itemized list of changes below, which I believe improved readability of the paper.

  1. The referee asks how do we know the dense set of singularities lies on the second sheet. It's an important concern. Addressing it was the main result of the paper, with details given in Sec. 3. It wouldn't have been possible to claim to have found a dense set of singularities unless they were all on the same sheet.

Summarizing the logic, analytic continuation of elastic unitarity guarantees that the singularities of the type studied in this paper have to lie on \emph{at least one} of the two sheets of the lightest-threshold branch cut in the variable $s$. Since it's known they do not appear on the first, they have to be located on the second one. These arguments by no means preclude the existence of ladder-type singularities on different sheets of other branch cuts.

  1. The referee proposes a mathematical construction for defining perturbative Feynman or time-ordered diagrams on other sheets, even if they are not related to the physical S-matrix by analytic continuation. As emphasized on page 21, the moment we commit to perturbation theory, there's really no obstruction to analytic continuation across any cut, as far as it's currently known, and hence also no puzzle of how to define perturbative S-matrix elements on different sheets. Performing such an analytic continuation in practice is a more complicated problem. (As an example, after plugging in a single Feynman diagram into Eq. (2.24), it truncates and gives an explicit formula for the analytic continuation on the second sheet of the lightest-threshold branch cut.) Still, I would like to emphasize that no Feynman diagrams appear in this paper.

It's also worth noting that the referee's proposal relies on the assumption that selecting $\pm i\varepsilon$'s for propagators is enough to explore all the sheets, or at least the ones that would have been otherwise disconnected in the full S-matrix. One can check on explicit examples, for instance the bubble or box diagrams, that this is not the case: the space of sheets is in general much larger than the space of choices of $i\varepsilon$'s. I would agree that \emph{certain} monodromies around singularities lying in the neighborhood of the physical kinematics can be computed this way.

---

## Round 2 · List of Changes

1. The referee asks about an example of a function with a natural boundary defined by its integral representation. Such examples were mentioned on page 3: elliptic functions (for example Jacobi theta or Dedekind functions) have a dense set of singularities for every rational value of the modular parameter $\tau$, forming a natural boundary. They are commonly defined by either infinite sum or product representations, differential equations, and integrals.

  2. As suggested by the referee, I added the clarification that the coefficient on the RHS of Eq. (2.11) is fixed by unitarity, but is immaterial to later discussion and hence chosen to be $4$ for convenience.

  3. Below Eq. (2.25), I clarified what is meant by assuming Mandelstam analyticity.

  4. This is a good catch. The sentence was meant to say "the analytic continuation of $\mathrm{Im}\, T(s,z)$".

  5. Corrected.

  6. That's right, there were two typos below Eq. (3.6).

  7. I agree. The advantage of using Eq. (1.1) is that it gives a necessary, not only sufficient, condition for anomalous thresholds, since the singularities on the RHS of the unitarity equation have to be also singularities of the LHS, so the sheet on which the singularity lies can be determined.

  8. I expanded the description of Fig. A.1, such that now all the information needed to understand it is contained in the caption, on top of the explanations already given in the main text.

  9. Corrected.

In addition to the above changes, I wasn't entirely happy with the presentation of the arguments leading to Eq. (2.10), so I rewrote the text around it to be more accessible.

---

## Editorial Decision

published